# Multi-System-Level Analysis with RNA-Seq on Pterygium Inflammation Discovers Association between Inflammatory Responses, Oxidative Stress, and Oxidative Phosphorylation

**DOI:** 10.3390/ijms25094789

**Published:** 2024-04-27

**Authors:** Ye-Ah Kim, Yueun Choi, Tae Gi Kim, Jisu Jeong, Sanghyeon Yu, Taeyoon Kim, Kisung Sheen, Yoonsung Lee, Taesoo Choi, Yong Hwan Park, Min Seok Kang, Man S. Kim

**Affiliations:** 1Translational-Transdisciplinary Research Center, Clinical Research Institute, Kyung Hee University Hospital at Gangdong, Kyung Hee University College of Medicine, Seoul 05278, Republic of Korea; yeak426@khu.ac.kr (Y.-A.K.); uag43@khu.ac.kr (Y.C.); symply501@khu.ac.kr (J.J.); sanghyeon99@khu.ac.kr (S.Y.); taeyuz@khu.ac.kr (T.K.); kisungsheen@khu.ac.kr (K.S.); ylee3699@khu.ac.kr (Y.L.); 2Department of Biomedical Science and Technology, Graduate School, Kyung Hee University, Seoul 02453, Republic of Korea; 3Department of Ophthalmology, Kyung Hee University Hospital at Gangdong, Kyung Hee University College of Medicine, Seoul 05278, Republic of Korea; tk1213@khu.ac.kr; 4Department of Urology, School of Medicine, Kyung Hee University, Seoul 05278, Republic of Korea; howdyhowdy@khu.ac.kr; 5Department of Microbiology, Ajou University School of Medicine, Suwon 16499, Republic of Korea; parky5@ajou.ac.kr; 6Department of Ophthalmology, Kyung Hee University Hospital, Kyung Hee University College of Medicine, Seoul 02447, Republic of Korea

**Keywords:** pterygium, inflammation, race, RNA sequencing

## Abstract

A pterygium is a common conjunctival degeneration and inflammatory condition. It grows onto the corneal surface or limbus, causing blurred vision and cosmetic issues. Ultraviolet is a well-known risk factor for the development of a pterygium, although its pathogenesis remains unclear, with only limited understanding of its hereditary basis. In this study, we collected RNA-seq from both pterygial tissues and conjunctival tissues (as controls) from six patients (a total of twelve biological samples) and retrieved publicly available data, including eight pterygium samples and eight controls. We investigated the intrinsic gene regulatory mechanisms closely linked to the inflammatory reactions of pterygiums and compared Asian (Korea) and the European (Germany) pterygiums using multiple analysis approaches from different perspectives. The increased expression of antioxidant genes in response to oxidative stress and DNA damage implies an association between these factors and pterygium development. Also, our comparative analysis revealed both similarities and differences between Asian and European pterygiums. The decrease in gene expressions involved in the three primary inflammatory signaling pathways—JAK/STAT, MAPK, and NF-kappa B signaling—suggests a connection between pathway dysfunction and pterygium development. We also observed relatively higher activity of autophagy and antioxidants in the Asian group, while the European group exhibited more pronounced stress responses against oxidative stress. These differences could potentially be necessitated by energy-associated pathways, specifically oxidative phosphorylation.

## 1. Introduction

A pterygium is a common condition where fibrous vascular tissue from the conjunctiva proliferates and grows onto the corneal surface or limbus. When a pterygium advances across the cornea, obstructing the visual axis or causing significant corneal astigmatism, resulting in blurred vision for the patient, surgical removal of the pterygium is performed [1]. Histologically, a pterygium is characterized by the proliferation of epithelial cells and fibroblasts, displaying features reminiscent of neoplastic tissue. A study has demonstrated the presence of TP53 tumor suppressor gene abnormalities in both limbal and pterygium cancers. A p53-dependent mechanism of damage-induced programmed cell death gradually acquires mutations in other genes. Pterygiums are also defined as tumor-altered limbal basal cells that, akin to other invasive cancers, secrete transforming growth factor-beta (TGF-β) and produce various matrix metalloproteinases (MMPs) [2]. The disruption of ocular-surface homeostasis occurs as the tissue from the conjunctiva and limbus invades the cornea in a wedge-shaped growth pattern. This leads to the occurrence of proliferative clusters of limbal stem cells, epithelial metaplasia, active fibrovascular tissue, and inflammation [3].

However, there is no clearly delineated pathologic pathway for the formation and progression of pterygiums, and our understanding of the role of heredity in this area remains limited. The primary prominent risk factor for the development of pterygiums is ultraviolet (UV) light exposure [4]. It can be speculated that genetic changes at the cellular level in normal conjunctival cells, along with alterations in the composition of tissue components due to these genetic variations, contribute to the formation of pterygiums. Therefore, analyzing genes that are either overexpressed or coordinately suppressed in pterygium tissue could potentially allow us to predict the underlying mechanisms behind pterygium development and assess the risk of recurrence after surgery. This understanding could aid in establishing effective strategies for the prevention and treatment of pterygiums.

Previous studies have identified potential functional genes associated with the pathogenesis or development of pterygium models [5,6,7,8] and have also underscored the connection between DNA damage, mutations in ion channel-associated genes, and the progression of specific ophthalmic conditions. Liu et al. [5] observed the significant downregulation of genes associated with oxidative stress response, including FOS, JUN, and DUSP1. Additionally, genes involved in ion transmembrane transport, such as CALM1, CALM2, and FXYD4, were significantly downregulated. Therefore, it is hypothesized that these genes may play a pivotal role in the exacerbation and progression of pterygiums. However, these studies have been applying global searches for analysis approaches, which are probably too broad to discover underlying functional genes effectively or efficiently. Furthermore, these studies have only reported on patients of a single ethnicity, failing to compare genetic differences across different racial groups.

In our pursuit of understanding the gene regulatory mechanisms closely linked to the inflammatory responses of pterygiums, we employed several analytical approaches. These ranged from conventional methods like gene set enrichment analysis (GSEA) to a machine learning-based approach called QLattice. Our study focused on two distinct ethnic groups: Asian and European pterygium patients. By applying these diverse methods, we aimed to identify the primary regulatory patterns of functional genes or gene modules shared or distinct between these two ethnic groups.

## 2. Results

### 2.1. Inflammation-Associated Differentially Expressed Genes Might Have Significant Roles

In our pursuit of elucidating the fundamental gene regulatory mechanisms associated with inflammatory responses in pterygiums, we meticulously analyzed and compared two distinct RNA-seq datasets. Initially, we applied batch correction and performed principal component analysis (PCA) on both the Asian and European datasets. After batch correction (as depicted in Figure 1A), we observed that the variations within all four groups—Asian pterygiums, European pterygiums, Asian control, and European control—were less than 20% for PC1 and 11% for PC2. The two European groups were positively correlated, and the other Asian groups were negatively correlated with each other within the reduced dimensions. A large number of significantly differentially expressed genes (DEGs) were visualized as volcano plots for both Asian and European pterygium samples. These plots demonstrate that the DEGs identified were sufficient for downstream analyses (Figure 1B). Of the DEGs, some appeared to be associated with pterygium development. For instance, (i) Il36g (IL-36γ), one of the pro-inflammatory agonists expressed by the corneal epithelium, was reported to be upregulated following corneal injury and was observed to be remarkably upregulated in our Asian pterygiums [9]. (ii) The expression of PAX6 was considerably downregulated in corneal epithelial cells isolated from patients with severe ocular surface inflammatory diseases such as Stevens–Johnson Syndrome or recurrent pterygiums [10]. (iii) Considering the impairment of autophagy as a potential cause of pterygiums, we observed that MALAT1, a long non-coding RNA that promotes autophagy in retinoblastoma cells, was markedly downregulated in the European pterygiums [11]. We examined inflammatory pathways significantly involved in differentially expressed genes (DEGs) by conducting the GO analysis using the curated set of inflammation-associated genes sourced from the MSigDB. From the top twenty common pathways shown in Figure 1C, some metabolic processes or biosynthetic processes were negatively enriched, while others, such as the regulation of binding or protein neddylation, were positively enriched. We also searched for differentially enriched inflammatory pathways between the Asian and the European pterygiums (Figure 1D). Among the European pterygium samples, the top pathways were associated with amide biosynthetic processes or mRNA metabolic processes. The top pathways in the Asian pterygium samples were, however, associated with the development of the nervous system, head, and brain. From another perspective, the inflammatory signaling cascade was upregulated in the Asian pterygiums, while the European pterygiums displayed upregulation in RNA processing and translational pathways.

### 2.2. GSEA Clusters Revealed Eye-Associated Inflammatory Responses

We additionally carried out GSEA to search for interactions between pathways that are known or at least linked in current databases. Pathway connectivity is visualized using Cytoscape in Figure 2A for the European pterygiums and Figure 2B for the Asian pterygiums. In the European pterygiums (Figure 2A), although inflammation-associated nodes involved in several diseases (other than eye-associated conditions) were positively and significantly enriched with straight lines (i.e., relatively lower *p*-values) such as otitis media (inflammation caused by an infection of the middle-ear) or colitis (inflammation in the colon), eye-associated inflammatory terms were also partially at least connected (i.e., relatively higher *p*-values).

While having inflammatory abnormality of the eye with a higher NES, inflammatory disorders in the uvea—such as uveitis, anterior uveitis, and iridocyclitis—as well as blepharitis and punctate keratitis (which share surface conditions similar to pterygiums) were positively enriched. However, a few other ophthalmological diseases, namely conjunctivitis and keratoconjunctivitis, whose transcriptional patterns appeared to be incomparable to those of pterygiums, were found to be downregulated. In the Asian pterygiums (Figure 2B), nodes directly connected to inflammatory abnormalities of the eye such as uveitis, anterior uveitis, and iridocyclitis were relatively more significantly and positively enriched with the straight lines. This may suggest that the inflammatory patterns observed in pterygiums may be similar to those seen in other ophthalmological diseases, and these patterns appeared more frequently in the Asian cases. Overall, the inflammatory patterns between the Asian and the European pterygiums were comparable and proportional, despite some unevenness and variation.

### 2.3. Core Inflammatory Genes of the Asian Pterygiums Demonstrated Significant Alterations on Inflammatory Responses and Oxidative Stress

To better understand how the transcriptional profiles of our Asian datasets were directly associated with inflammatory responses, we explored the relative gene expression levels of the customized inflammatory gene lists obtained from [12] (Figure 3). The gene list is composed of six different categories, where three of them (i.e., innate immunity, extracellular immunity, and mitochondrial innate immunity) are typical systems intimately associated with inflammatory responses, and the other three (i.e., integrated stress responses, renin–angiotensin–aldosterone system, and unfolded protein responses) are additional systems indirectly involved in cellular stresses. As initially designed in this study, we primarily focused on gene expression profile alterations associated with various inflammatory responses. Among the list of genes with significant up/downregulations in Asian pterygiums, the eye-associated inflammatory gene list included (i) the upregulation of CSF3, a cytokine linked to increased pain perception in the contralateral eye post-cataract surgery, which represented its role in inflammatory responses [13]; (ii) the upregulation of SOCS1, which indicated an association with modulating ocular inflammation, as observed in intraocular inflammatory diseases like uveitis and scleritis [14]; (iii) the upregulation of HLA-A in ocular diseases, which demonstrated its role in increasing the risk of ocular lesions in Behçet’s disease [15]; (iv) the upregulation of CD38, which corresponds with findings from retinal ischemia/reperfusion injury studies, where a deficiency in CD38 was shown to confer protective effects against inflammatory responses on the retina [16]; (v) the upregulation of JUN in glaucoma, which displayed its involvement in retinal ganglion cell (RGC) degeneration through the JNK-JUN signaling pathway [17]; (vi) the upregulation of PYCARD, coupled with the downregulation of NLRP3, which highlighted the significance of the NLRP3 inflammasome and PYCARD in IL-1β production in the eye [18]; (vii) the upregulation of TNFRSF10B and downregulation of TNFSF10, which may express a complex interplay in retinal degenerative disorders of age-related macular degeneration (AMD), where TNFSF10 and its receptor TNFRSF10B are involved in inflammatory and apoptotic pathways [19]; (viii) the upregulation of SERPING1, whose genetic variations can be significant contributors to AMD susceptibility [20]; (ix) the upregulation of CXCL8, alongside the downregulation of IL6, known to be associated with inflammatory responses mediated by CXCR1 and CXCR2 receptors [21], which turned out to be important in ocular inflammation and angiogenesis [22]; (x) the downregulation of CXCL11, involved in the inflammatory processes of AMD, which was found to be linked to the retinal pigment epithelium’s (RPE’s) responses [23].

While having found a crucial association between our pterygiums’ transcriptional profiles and the core inflammatory genes, we also looked into how significantly inflammation-associated genes in the pterygium are significantly involved in stress responses. The inflammatory gene list with significant alterations associated with stress responses includes the following: (i) The downregulation of ZBP1 whose activation is necessitated by oxidative damage could exhibit involvement in oxidative stress-induced inflammatory signaling [24]; (ii) the downregulation of GCLC and GPX2 could correlate with activation of the Nrf2 pathway in oxidative stress responses in ocular diseases [25]; (iii) the downregulation of LONP1 in association with CODAS syndrome suggests that the suppression of LONP1 may play a significant role in mitochondrial dysfunction [26]; (iv) the downregulation of FOXO3, an indicator of oxidative stress and cell homeostasis, might be associated with an impaired response to oxidative stress [27]; (v) the downregulation of SOD2 implies a possible connection to mitochondrial dysfunction or its oxidative stress in RPE cells that can lead to metabolic changes and structural damage in photoreceptors [28]; (vi) the upregulation of ATG12 and ATG16L1, which are autophagy markers, may strengthen the role of autophagic processes in disease developments [29]; (vii) the upregulation of DNAJC3 and downregulation of ATF6, involved in complex dynamics of endoplasmic reticulum (ER) stress and unfolded protein response (UPR) signaling pathways in ocular health, might suggest probable ER stress or disruption in UPR-mediated cellular responses [27].

### 2.4. Inflammatory Pathways Displayed Dynamic Alterations on Asian Pterygiums

To estimate how the activation levels of the inflammatory pathways revealed dynamic alterations in Asian pterygiums, we computed enrichment scores of each pathway in the Asian cases compared to the non-Asian cases. The corresponding normalized enrichment scores (NESs) were acquired through fast gene set enrichment analysis (fGSEA) [30] using the aforementioned customized inflammatory modules.

As shown in Figure 4, the innate immune system, which encompasses well-known interferon-driven and interferon-stimulated genes (ISGs), was downregulated in the Asian pterygiums, except for antigen presentation. Both canonical and non-canonical modules of the innate immune system, including most of the ISGs, may not only play a role in the early stages of the innate immune responses against invading pathogens using reactive oxygen species (ROS) [31], whereas they exhibit downregulation in inflammatory diseases such as HIV [32]. 

In the extracellular immunity system, the positive activation of cell surface markers and surface receptor signaling in Asian pterygiums could potentially necessitate the adaptation of inflammatory responses such as immune privilege [34]. While cytokines (i.e., CXC chemokine receptors) were also activated, interleukins were suppressed in the pterygium. A deficiency of interleukins or an imbalance of cytokines is known to often occur in autoimmune diseases [35,36]. 

Considering the mitochondria innate immune system, we estimated enrichment score changes for modules such as mtDNA, dsRNA, and mtdsRNA. Mitochondrial ROS (mROS), which can be generated by antiviral signaling and are also remarkably associated with oxidative stress, could potentially result in inflammatory disorders [37]. 

Alterations in unfolded protein response (UPR), which are important for the functions of immune cells, innate immune signaling, and oxidative stress within the endoplasmic reticulum [38], have been reported to be involved in various diseases, including cancer and inflammatory bowel disease [39].

The renin–angiotensin–aldosterone system (RAAS) plays a critical role in inflammation and oxidative pathways. Abnormalities in the angiotensin regulatory axis could cause tissue damage, inflammation, and NADPH activation, leading to oxidative stress. The imbalance of RAAS is also well known in several disorders associated with inflammation, such as obesity and cardiovascular disease [40]. 

The integrated stress response (ISR), which is intimately associated with oxidative stress, has demonstrated positive activations of antioxidants, autophagy, ISR activators, and cytokines/chemokines. This may suggest that the repair and rebuilding system, which addresses damages caused by oxidative stress and inflammatory responses, can be revitalized.

### 2.5. Compatible Patterns between the Asian and European Pterygiums Are Associated with Oxidative Stress

To search more extensively for inflammatory-based regulations of genes or modules, we compared alteration patterns of the core genes or their modules (i.e., gene groups) between the Asian and the European pterygiums. As depicted in Figure 4 and Figure 5, the inflammatory pathways with the most similar patterns of gene expression alterations between the two ethnic groups turned out to be extracellular immunity and UPR. In these pathways, cytokines, surface marker receptor signaling, and the endoplasmic reticulum were positively enriched, while antigen presentation, interleukins, and mitochondrial modules were negatively enriched. RAAS and ISR displayed significantly enriched alterations in opposite patterns between the two datasets. Specifically, bradykinin production, complement activation/fibrin deposition, and hyaluronan accumulation in RAAS and death factors in ISR demonstrated contrasting enrichment. 

Despite recognizing these common alterations or contradictory patterns between the Asian and the European pterygiums, we continuously focused on associations between cellular stresses and inflammatory responses, given that oxidative stress already plays pivotal roles in immune-associated activities [41]. The following compatible patterns of some representative inflammatory genes between the two groups were observed: (i) The downregulation of ZBP1 in both pterygiums, known for its involvement in antiviral immunity, could be associated with cellular stress, including oxidative stress. RPE (retinal pigment epithelial) cells exposed to chronic low-level oxidative stress could induce damage to mtDNA, leading to subsequent translocation of the damaged DNA, resulting in the binding and activation of ZBP1 [24]. (ii) The upregulation of FAS in both pterygiums, known as an important cell surface receptor initiating cell apoptosis, could be a consequence of increased oxidative stress. Oxidative stress generally increases the overall expression of both FAS and FAS-L genes, and a high level of ROS is involved in the upregulation of two genes [42]. (iii) The downregulation of the GPX family (GPX1 as a cofactor to shield cells from oxidative stress, GPX2 as an electron donor to convert harmful ROS, and GPX4 as a cofactor to convert lipid hydroperoxides into lipid alcohols), inherently linked to oxidative stress, might deactivate its defensive mechanism against oxidative stress [43].

### 2.6. Oxidative Phosphorylation Might Cause the Different Responses against Oxidative Stress in European Pterygiums

In an effort to uncover shared and distinctive patterns in the Asian and the European pterygiums, protein–protein interaction (PPI) analysis through STRING revealed enriched pathways in both the Asian and the European pterygiums compared to the controls. By examining differentially expressed inflammatory genes, we identified approximately 500 enriched pathways from the biological processes of Gene Ontology in both the Asian and the European cases. Among them, multiple enriched pathways were shared in both ethnic groups, including immune pathways, tissue remodeling, TGF beta-related pathways, and stress response in both Asian and European pterygiums. Note that the core inflammatory genes displayed the upregulation of genes associated with oxidative stress response in both types of pterygiums (Figure 4 and Figure 5), and the PPI network also unveiled enrichment in stress response in both cases. However, the PPI network of European pterygiums exhibited significant alterations in oxidative phosphorylation that were absent in Asian pterygiums. This finding was further detected in lollipop plots, indicating an overall upregulation of oxidative phosphorylation in the European and a contrasting downregulation in the Asian pterygiums (Figure 6B). In all complexes which were associated with the electron transport chain, there were more upregulated genes in the European group than in the Asian pterygium group (Figure 6C). Therefore, the heatmap of oxidative phosphorylation consistently supported these findings, reinforcing the upregulation of oxidative phosphorylation in pterygiums of the Europeans, observed in the aforementioned analysis.

### 2.7. Correlated Inflammatory Modules Might Play an Important Role

In attempts to explore correlation patterns between functional modules of inflammation-associated systems, we implemented a custom-made method adopted from QLattice [30], a machine learning-based algorithm for omics data, where we applied normalized enrichment score (NES) and Pearsons’ correlations (Figure 7). The Pearson’s correlation coefficient for each functional module (or sub-system) was obtained through the QLattice regression model with NESs of the modules within the corresponding system as inputs. Considering our comparative analysis on the Asian and European datasets, the inflammation module of innate immunity and the endoplasmic reticulum of UPR exhibited relatively higher correlations (i.e., higher than 0.6) with the modules of their corresponding systems (i.e., pathways) in both Asian and European samples. This suggests that the genes (Table 1) involved in those modules might be coordinately regulated regarding either the innate immune system or the unfolded protein response in the pterygiums of both ethnic groups. 

The correlation coefficients of some modules such as interleukins of extracellular immunity, mtDNA of mitochondrial innate immunity, mitochondrial modules of the unfolded protein response, ISR activators of the integrated stress response, and NADPH oxidase of the renin–angiotensin–aldosterone system exceeded 0.6 only in Asian samples. European pterygiums also displayed a few modules (surface marker/receptor signaling and cytokines in extracellular immunity) whose coefficients were higher than 0.6 only in European samples. This may indicate a coordinated regulation of genes of each group associated with their corresponding modules in either Asian or European pterygiums. However, some modules demonstrated very low correlation coefficients in at least one of the groups and also substantial differences between them. These modules were as follows: (i) mtDNA/dsRNA of mitochondrial innate immunity, antioxidants of the integrated stress response, and syndecans of the renin–angiotensin–aldosterone system in the Asian pterygiums; (ii) interleukins of extracellular immunity, mtdsRNA of mitochondrial innate immunity, mitochondrial modules of the unfolded protein response, and death factors of integrated stress response in the European pterygiums. These aforementioned correlations between the modules could possibly explain which modules seemed to be associated with potential primary patterns of pterygium inflammation or differential inflammation patterns between the Asian and the European pterygiums.

## 3. Discussion

In this study, we investigated gene regulatory mechanisms, specifically focusing on inflammation and its corresponding stress (i.e., oxidative stress) in pterygiums, utilizing RNA-seq datasets from pterygium patients, including both Asian and European cohorts. Our analyses on the pterygium datasets captured biologically and computationally (mathematically) sensible transcriptional profile alterations in both pterygiums compared to their controls and the comparison between the Asian and the European pterygiums. The comprehensive analysis of pterygiums encompassed identifying signaling pathways associated with existing databases through Gene Ontology (GO) analysis and gene set enrichment analysis (GSEA). Additionally, we investigated potential regulatory changes in the core inflammatory gene list, assessed activity levels of essential inflammatory functional modules, and computed correlations between these functional modules.

Although typical analyses (i.e., GO and GSEA) on the Asian and the European pterygium data displayed eye-associated dysfunctions such as inflammatory abnormality of the eye from GSEA, our investigation further dipped into underlying potential regulatory mechanisms through the core inflammation-associated genes and their modules. Among commonly upregulated inflammatory functional modules (cytokines and surface marker receptor signaling in extracellular immunity; endoplasmic reticulum in UPR; NADPH oxidase in RAAS, and antioxidants, cytokines/chemokines and ISR activators in ISR), the endoplasmic reticulum in UPR showed consistency with the upregulation of ‘retrograde vesicle-mediated transport, Golgi to endoplasmic reticulum’ identified by pathway enrichment analysis with GO. Also, three modules of ISR were upregulated in both cases, suggesting that stress responses are associated with inflammation in pterygiums since ISR is expected to deal with various stressors, including proteostasis defects, nutrient deprivation, viral infection, and redox imbalance. ISR, closely linked to NF-κB, facilitates the transcription of proinflammatory genes. Upon activation, ISR also triggers the secretion of inflammatory cytokines, promoting cell communication during local inflammation. Among the frequently upregulated genes within these modules, GSTA1 plays a crucial role in detoxifying oxidative stress products [50]. PARL in the antioxidant of ISR detoxified lipid peroxidation and inhibited ferroptosis [12]. FOS in cytokines/chemokines of ISR was associated with the DNA damage response [51]. As the NESs of the antioxidant and cytokines/chemokines indicated their positive enrichment in pterygiums in both Asian and European samples, GSTA1, PARL, and FOS might contribute to their upregulation in response to oxidative stress and DNA damage.

Concerning commonly downregulated inflammatory pathways, canonical and non-canonical innate immunity, antigen presentation and interleukins in extracellular immunity, mtDNA/dsRNA in mitochondrial innate immunity, mitochondrial UPR, AGT regulator axis and PANoptosis in RAAS, and AA-uptake/biosynthesis and survival factors in ISR (Figure 4 and Figure 6) were included. Many of the genes in these negatively enriched modules were involved in well-known inflammatory signaling pathways: (i) the JAK/STAT signaling pathway, (ii) the MAPK signaling pathway, and (iii) the NF-kappa B pathway. The three pathways are regarded as the main inflammatory pathways that cooperatively interact with each other [66]. Among the downregulated genes in pterygium, CXCL11, ADAR, XAF1, ZBP1, HERC5, CIITA, IL6, IL10, IL12RB2, IRF3, CGAS, SOD2, and CMA1 have been found to be directly or indirectly associated with JAK/STAT signaling pathways [67,68,69,70,71,72,73,74,75,76,77,78,79,80]. Several genes were also involved in MAPK signaling pathways, including CXCL11, ADAR, PLVAP, PARP10, XAF1, ZC3HAV1, ZBP1, F12, CIITA, IL6, IL10, IRF3, CGAS, LONP1, FOXO3, and SOD2 [81,82,83,84,85,86,87,88,89,90,91,92,93,94,95,96]. Additionally, the following genes contributed to NF-kB pathways in either a negative or positive way: CXCL11, ADAR, LGALS3BP, PARP10, ZBP1, HERC5, CIITA, IL6, IL10, RELB, NFKB2, CGAS, and SOD2 [78,97,98,99,100,101,102,103,104]. Intriguingly, CXCL11, ADAR, ZBP1, CIITA, IL6, IL10, CGAS, and SOD2, which were referred to as being associated with all three inflammatory signaling pathways, were all downregulated in both Asian and European pterygium cases. The JAK/STAT, MAPK, and NF-kB signaling pathways also play roles in oxidative stress and DNA damage, where increased oxidative stress impairs IFN-alpha signaling, affecting JAK/STAT and the inactivation of JAK1, while oxidative stress and DNA damage activate MAPK and NF-kB [105,106,107]. Since the pathway enrichment analysis showed a stress-activated inflammatory signaling cascade in the Asian pterygium, the downregulation of the genes associated with the inflammatory signaling pathways could be expected. It would appear, therefore, that both positive and negative changes in inflammatory signaling pathways in response to oxidative stress and DNA damage in pterygiums could be considered.

Inconsistent alteration patterns between the two pterygium groups, upregulated in the Asian and downregulated in the European pterygiums, converged into autophagy of the immune system. According to previous studies, autophagy could affect MHC class 1 presentation in which HLA-B played a role [108] and negatively regulated MAVS signaling [109]. Also, BECN1, ATG16L1, BAK1, TNFRSF10B, and C1QA might contribute to inflammatory response and autophagy [44,110,111,112,113]. Another contradictory pattern, downregulation in the Asian and upregulation in the European pterygiums, represented different pathogeneses of pterygiums between the two groups. For instance, CASP1 could participate in pyroptosis through activated IL1B and IL18, and FASLG could be involved in pterygium fibroblasts, suggesting that they might play a role in pterygium pathogenesis [64,114]. Therefore, there might be different responses between the Asian and the European pterygiums with regard to autophagy or pathogenesis.

The PPI network revealed common enriched pathways, potentially linked to the pathology of pterygiums, including inflammatory responses, the regulation of leukocyte differentiation, tissue remodeling, the TGF beta receptor signaling pathway, the positive regulation of superoxide anion generation, and the ROS metabolic process. Notably, TGFB1 emerged as a central node in the network across both the Asian and the European groups, contributing to all enriched pathways: immune pathways, tissue remodeling, TGF beta signaling pathways, and stress response. The role of TGFB1 in tissue remodeling, particularly its interaction with fibroblasts, is implicated in pterygium tissue remodeling [115]. Given the association of TGFB1 with inflammation and its pro- and anti-inflammatory responses [116], it likely plays multi-functional roles in pterygium pathology, encompassing inflammation, stress response, and tissue remodeling.

In contrast, an additional dissimilarity between the Asian and the European pterygium groups emerged in PPI analysis, specifically involving oxidative phosphorylation. While the Asian and the European pterygiums shared general patterns in inflammation, tissue remodeling, and stress response, oxidative phosphorylation was significantly enriched only in the European pterygiums. Furthermore, the lollipop plot and the heatmap illustrated an overall upregulation of oxidative phosphorylation in the European pterygiums, particularly with the significant downregulation of complex IV in the Asian samples and its upregulation in the European samples (Figure 6B,C). A previous study also found differences in oxidative phosphorylation between Caucasian and Hispanic pterygiums [117]. Consequently, ethnic variations in pterygium pathology demonstrated significant differences in energy-associated pathways, specifically oxidative phosphorylation. This observation could contribute to the inconsistent patterns observed in the Asian and the European pterygium pathogeneses.

The limitation of this study lies in its lack of consideration for clinical aspects. Although this study primarily focused on deciphering inflammatory patterns detectable during the progression of pterygiums from a basic research perspective, directly applying the findings to clinical practice may pose challenges. Nevertheless, even so, when contemplating clinical treatment for pterygium-associated inflammation, treatment may not be required in the early stages of a pterygium. However, topical antibiotic therapy could be applied to suppress pterygium inflammation or progression. Additionally, in cases where the progression is rapid, it is believed that the use of anti-inflammatory agents such as vasoconstrictors and steroids can slow down its advancement.

## 4. Materials and Methods

### 4.1. Patient Samples

In our sample collection, we obtained pterygium tissues from Korean patients, which were categorized as Asian pterygiums for comparison with the other data categorized as European pterygiums. The sample collection was approved by the Institutional Review Board of Kyung Hee University Hospital at Gangdong (IRB No. 2022-04-006) and was conducted in accordance with the official regulations for clinical research and the Declaration of Helsinki. The Asian cases consisted of six patients diagnosed with primary pterygiums who underwent elective pterygium excision. These patients provided written informed consent. Excluded from the study were patients with recurrent pterygiums, conjunctival tumors, or ocular surface inflammation. All Asian patients were in good health and ranged in age from 61 to 72 years, including three males and three females. Conventional excision of pterygiums was performed under local anesthesia. Whole pterygial tissues, including pterygial heads and bodies, were collected from the nasal limbus, and a small section of conjunctival tissue was collected as a control. Finally, the tissues were transferred to 2 mL Eppendorf tubes (Eppendorf, Hamburg, Germany) and stored at −80 °C for subsequent total RNA extraction. Therefore, we obtained six pterygial tissues and six conjunctival tissues, where each patient contributed both a pterygial and a conjunctival sample. Also, we downloaded publicly available data from the Gene Expression Omnibus (GEO), whose accession number was GSE155776 [118], and considered these data as the European pterygiums. The European data included eight human conjunctival pterygium samples and eight healthy conjunctival samples as controls. Their clinical details pertaining to the study patients are described in Table 2.

### 4.2. RNA-Seq Preparation and Preprocessing

For the Asian data, gene expression levels were obtained for 12 samples, consisting of 6 independent libraries for the pterygiums and their corresponding controls. A quality control process including FASTQC (v0.11.7) was performed on the sequencing data before aligning the reads. The STAR-HTSeq workflow, containing STAR (v2.7.3a) [119] and HTSeq-count (v0.12.4) [120], was used to map the sequencing reads to the reference genome (GRCH38) along with its annotation. After having the read counts, gene expression levels were normalized and differentially analyzed using the DESeq2 package [121], with normalization as VST (variance stabilizing transformation).

### 4.3. Differential and Enrichment Analysis

ComBat batch correction [122] in the sva package [123] was executed to mitigate bias in the data from the European and the Asian samples independently. While recognizing ethnic differences and distinct sample origins as potential batch effects, we performed batch correction with European and Asian designations as batch variables. Principal component analysis (PCA) was implicated with regularized-logarithm transformation and PCA function within the DESeq2 R package (ver. 1.38.3). Also, the differential analysis of ‘Pterygium vs. Control’ using inflammation-associated genes of both groups was carried out through DESeq2, and volcano plots depicting differentially expressed genes were generated by EnhancedVolcano (ver. 1.16.0) [124]. Additionally, enriched functional pathways for either the Asian or the European pterygiums were searched through Gene Ontology (GO) analysis using enrichGO from the ClusterProfiler R package (ver. 4.6.2) [125], and associations between enriched functional pathways were captured by gene set enrichment analysis (GSEA) (ver. 4.3.2) [126], followed by its visualization through Cytoscape, an open-source platform to visualize networks, along with EnrichmentMap, a visualization tool specifically designed for enrichment analysis [127]. A protein–protein interaction (PPI) network was illustrated using STRING, which provides potential associations between proteins in network format, utilizing differentially expressed inflammatory genes [128]. In the analyses (i.e., Volcano, GO, GSEA, and PPI), a gene list intimately associated with inflammation was selectively applied as acquired from the Molecular Signatures Database (MSigDB) [33]. For this gene list, we curated a list by collecting genes associated with a biological process (BP) of a category, ontology gene sets, focusing on homo sapiens for source species with all contributors, while looking up “inflammation” as a keyword on the MSigDB. For further analyses such as gene expression profile heatmaps and inflammatory-specific pathway enrichment analysis, the custom-made core inflammatory gene list and their corresponding inflammatory pathways and modules were adopted from [12], and also, fast gene set enrichment analysis (fGSEA) [129] was implemented. Additionally, MitoCarta 3.0 was applied to visualize patterns of oxidative phosphorylation conducted by fGSEA, while customized mitochondrial genes obtained from Guarnieri et al. (2023) were utilized for gene expression profile heatmaps related to oxidative phosphorylation [130,131].

### 4.4. Correlation Analysis between Inflammatory-Modules

QLattice [30], a machine learning-based regression and classification tool, was used to conduct correlation analysis between inflammatory pathways and their modules. This analysis displayed the top ten regression models while applying Pearson’s correlation. The computation utilized normalized enrichment scores (NESs) of inflammatory pathways and their modules between pterygiums and controls for the Asian and the European samples. After computing NESs of the inflammatory pathways compared to the controls for each pterygium sample, and for all pterygium samples compared to the controls in each ethnic group, we used the NESs of the inflammatory pathway as output and created a model with the NESs of its modules as input.

## 5. Conclusions

In conclusion, the inflammatory pathway played a crucial role in both the Asian and the European pterygiums, exhibiting commonalities and differences. The upregulation of antioxidant genes in response to oxidative stress and DNA damage suggested the relevance of oxidative stress and DNA damage to pterygium progression. The downregulation of the genes participating in the three major inflammatory signaling pathways, the JAK/STAT, MAPK, and NF-kappa B signaling pathways, indicated that the impairment of these pathways would be associated with pterygiums. Moreover, autophagy and antioxidants were relatively more active in the Asian pterygiums, while relatively more active stress responses against oxidative stress were observed in the European pterygiums. OxPhos could potentially be a source of these differences.

## Figures and Tables

**Figure 1 ijms-25-04789-f001:**
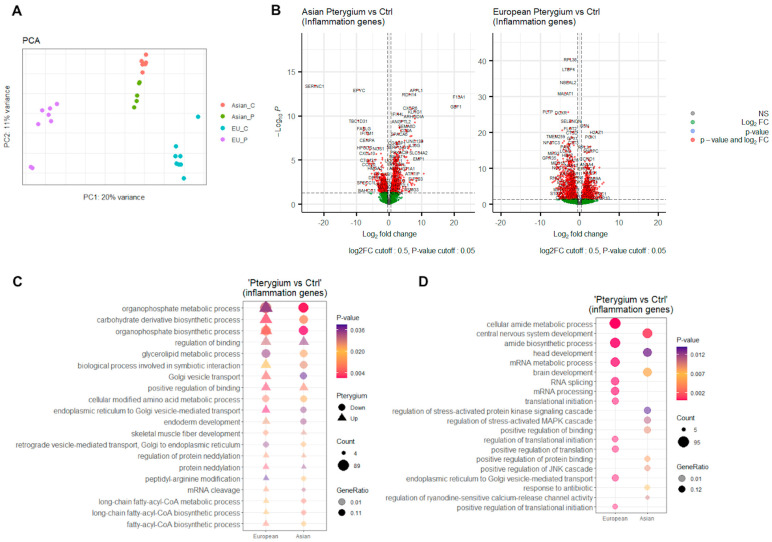
PCA result, differentially expressed genes, and enriched pathways. (**A**) PCA result visualized by PC1 and PC2 for pterygium groups and control groups in the Asian and the European samples using inflammatory-associated genes from MSigDB. (**B**) Differentially expressed inflammatory-associated genes in the Asian and the European pterygiums compared to the controls (*p*-value < 0.05, and log2 fold-change > 0.5). (**C**) Top 20 most commonly enriched pathways in the Asian and the European pterygiums compared to the controls (*p*-value < 0.05, and log2 fold-change > 0.25). (**D**) Top 10 most differentially enriched pathways in the Asian and the European pterygiums compared to the controls (*p*-value < 0.05, and log2 fold-change > 0.25).

**Figure 2 ijms-25-04789-f002:**
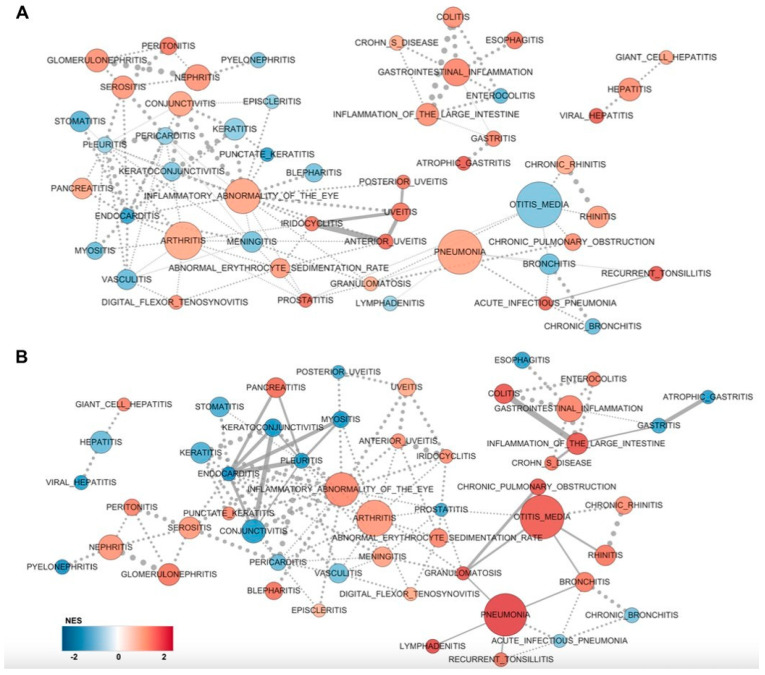
Enriched pathways network identified by GSEA. (**A**) Network visualization of enriched pathways from the European pterygiums compared with the control. (**B**) Network visualization of enriched pathways from the Asian pterygiums compared with the control. Enriched pathways in the pterygiums are depicted in red, while pathways enriched in the control are shown in blue. The size of nodes corresponds to the size of the gene set belonging to the respective pathway. Edges (similarity < 0.375) are generally represented as dotted lines, with particularly significant connections (*p*-value < 0.25) displayed as solid lines. Since edges represent the similarity coefficient between connected nodes, thicker lines indicate a higher degree of association.

**Figure 3 ijms-25-04789-f003:**
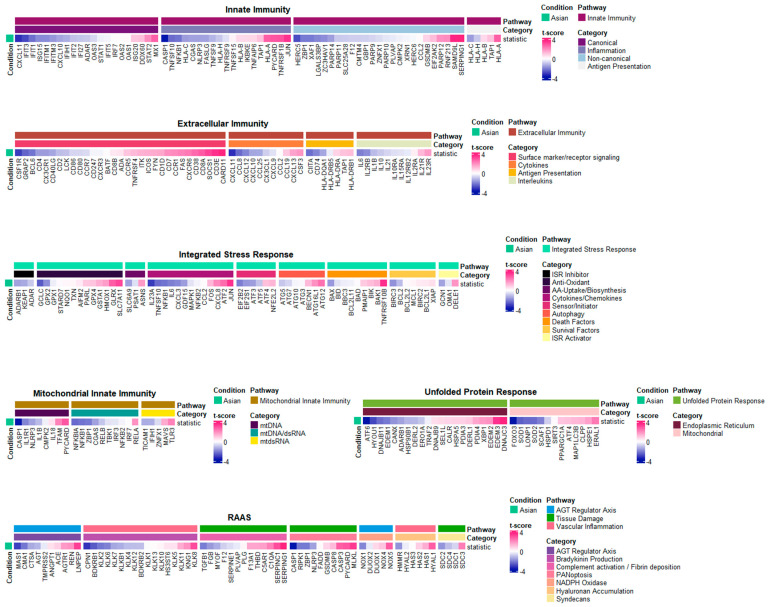
Heatmap of genes associated with inflammatory pathways with t-score comparing the Asian pterygiums and the control. Transcriptional profile alterations (as t-scores) of the inflammatory genes selectively organized by Topper and Guarnieri et al. (2023) [12] were visualized. Red indicates upregulation and blue indicates downregulation in either the Asian or the European pterygiums compared to the controls.

**Figure 4 ijms-25-04789-f004:**
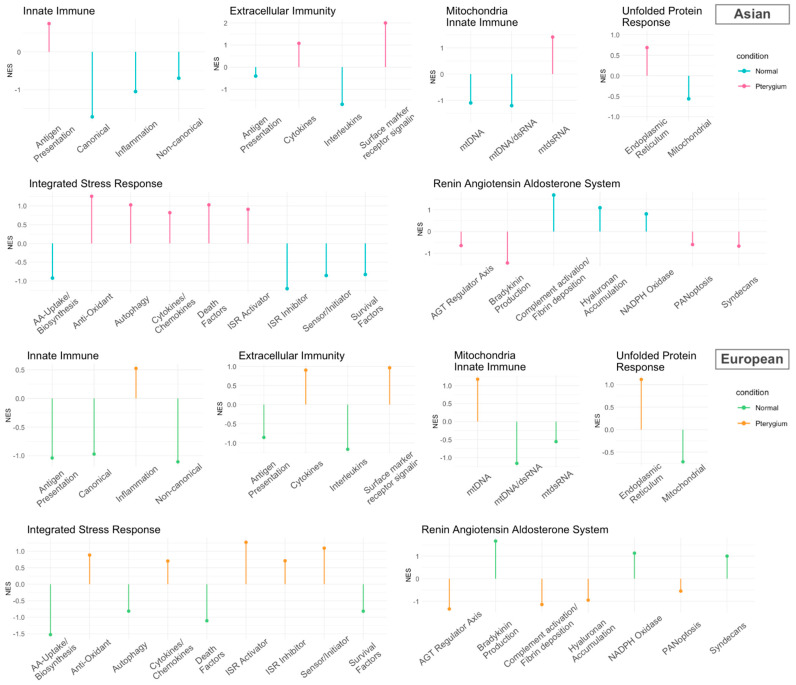
Lollipop plot using fGSEA with inflammatory genes customized by Topper and Guarnieri et al. (2023) [33] in the Asian and European pterygium groups compared to their control groups. The NESs were computed with the core inflammatory genes (*p*-value < 0.5).

**Figure 5 ijms-25-04789-f005:**
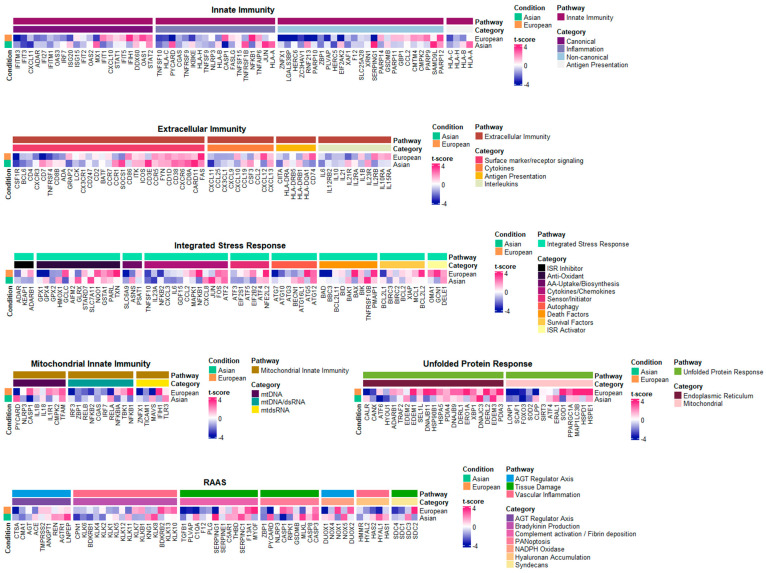
Heatmap of genes associated with inflammatory pathways with t-score, comparing the Asian pterygiums and the European pterygiums. Transcriptional profile alterations (as t-scores) of the inflammatory genes selectively arranged by Topper and Guarnieri et al. (2023) [33] were visualized. Red indicates upregulation and blue indicates downregulation in either the Asian or the European pterygiums.

**Figure 6 ijms-25-04789-f006:**
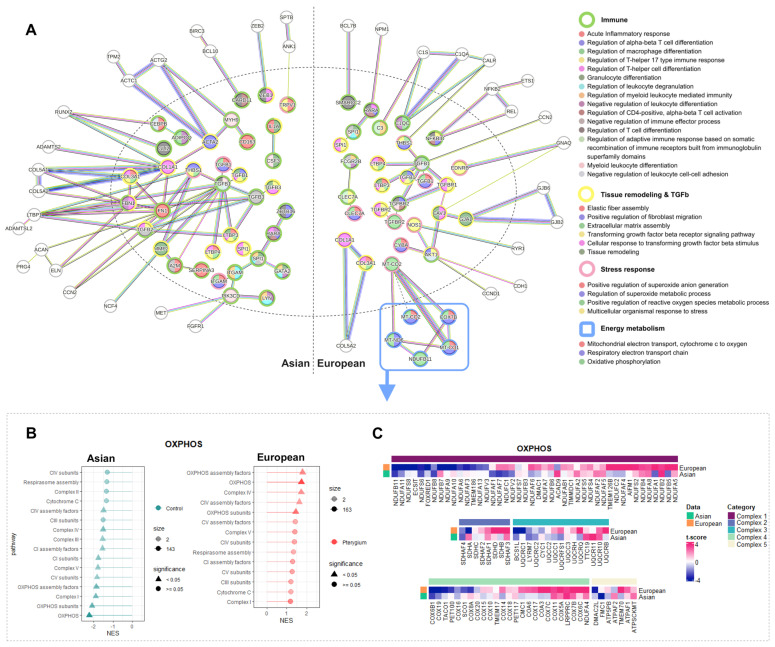
Functional differences with different activity levels in Asian and European pterygiums compared to their controls. (**A**) PPI network with significantly differentially expressed inflammatory genes. (**B**) Lollipop plot of oxidative phosphorylation. (**C**) Heatmap of oxidative phosphorylation. The inflammatory genes from MSigDB were filtered with adjusted *p*-value < 0.2 and |log2 fold change| > 0.5 for Asian, and adjusted *p*-value < 0.001 and |log2 fold change| > 2.25 for European. The proteins in the PPI network were exhibited by applying confidence 0.9. The enriched pathways were selected with strength 0.75 for Gene Ontology biological processes.

**Figure 7 ijms-25-04789-f007:**
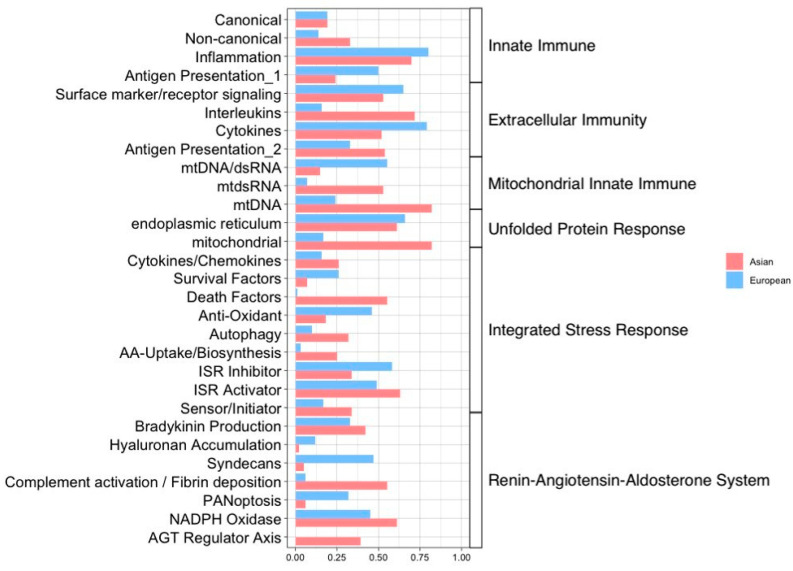
Pearson’s correlations of modules of the inflammatory pathways. The correlations computed by Q-Lattice with NES values of modules and their inflammatory pathways were visualized. The NES was calculated with the core genes of both the Asian pterygiums (*p*-value < 0.35) and the European pterygiums (*p*-value < 0.3). Pink indicates the Asian and sky blue indicates the European samples.

**Table 1 ijms-25-04789-t001:** List of core inflammatory genes associated with pterygiums.

Asian (Korean)	European (Germany)	Inflammation	Eye Disease or Pterygium Pathogenesis	Oxidative Stress or DNA Damage
up	up	**EI**; FYN, CD1D, CD38, CXCR6, CD8A, FAS **ISR**; GSTA1, FOS, ATF2, DELE1 **UPR**; PDIA4, XBP1, EDEM3	FYN [44], CD1D [45], ATF2 [46], XBP1 [47]	CD38 [48], FAS [49], GSTA1 [50], FOS [51], DELE1 [52], XBP1 [53]
down	down	**II**; IFITM3, IFIT3, LGALS3BP, ZC3HAV1, ZBP1 **ISR**; SLC6A9 **MII**; ZBP1 **UPR**; LONP1, SCAF1, FOXO3, SOD2 **RAAS**; CTSA, CMA1, ZBP1	LGALS3BP [54], FOXO3 [55], SOD2 [56], CMA1 [57]	ZBP1 [24], SCAF1 [58], FOXO3 [59], SOD2 [28], CTSA [60], CMA1 [61]
up	down	**MII**; MAVS **RAAS**; C1QA, SERPING1, HYAL1	C1QA [62], SERPING1 [63]	
down	up	**II**; CASP1, NFKB1 **MII**; CASP1, IL1R1 **ISR**; ADARB1, EIF2B2 **RAAS**; SDC2	CASP1 [64]	CASP1 [65]

**Table 2 ijms-25-04789-t002:** Clinical and demographic information for patients.

Case NO	Age Mean (std)	Sex	Tissue (Diagnosis)	Race
1	61	F	Primary pterygium/healthy conjunctiva	Asian (Korea)
2	65	F	Primary pterygium/healthy conjunctiva	Asian (Korea)
3	61	M	Primary pterygium/healthy conjunctiva	Asian (Korea)
4	72	M	Primary pterygium/healthy conjunctiva	Asian (Korea)
5	72	M	Primary pterygium/healthy conjunctiva	Asian (Korea)
6	64	F	Primary pterygium/healthy conjunctiva	Asian (Korea)
7–14	57.6 (8.5)	6/2 (M/F)	Primary pterygium	European (Germany)
15–22	55.8 (7.9)	6/2 (M/F)	Healthy conjunctiva	European (Germany)

## Data Availability

The data presented in this study are available on request from the corresponding author.

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
