# Peer review of "Multi-System-Level Analysis with RNA-Seq on Pterygium Inflammation Discovers Association between Inflammatory Responses, Oxidative Stress, and Oxidative Phosphorylation"

_ijms, 2024, doi:10.3390/ijms25094789_

Round 1
Reviewer 1 Report
Comments and Suggestions for Authors
This is an interesting study. The authors collected RNA-seq from pterygial and conjunctival tissues (as controls) from six patients, and retrieved publicly available data, including eight pterygium samples and eight controls. They investigated the intrinsic gene regulatory mechanisms closely linked to the inflammatory reactions of pterygium and compared results from Asian (Korea) and European (Germany) patients. The analyses revealed transcriptional profile alterations in both pterygium groups compared to controls, as well as differences between Asian and European cohorts. Various analytical approaches were employed, including GO analysis and GSEA, to identify signaling pathways and potential gene regulation changes.
However, several corrections should be considered:
1. The introduction of the article seems well-structured and informative. However, there are a few minor grammatical and stylistic suggestions to enhance clarity and readability:
1.1. Introduction: Delete "a" from second sentence in "a pterygium".
1.2. Line 48: In the third sentence, consider rephrasing "exhibiting characteristics similar to neoplastic tissue" for clarity and specificity.
2. Regarding Material and Methods section:
2.1. In the first paragraph, consider rephrasing "For our data considered to be the Asian pterygium" to improve clarity.
2.2. Specify the method of patient recruitment and inclusion/exclusion criteria.
2.3. Provide more information on the tools and databases used for differential and enrichment analysis, such as EnhancedVolcano, Gene Ontology (GO) analysis, GSEA, and STRING.
3. Results section:
3.1. Consider rephrasing "In line with our aim of discovering the primary gene regulatory mechanisms underlying inflammation in pterygium" to improve clarity and conciseness.
3.2. Provide more details on the methodology used for batch correction and principal component analysis (PCA),
3.3. Provide more context on the customized inflammatory gene lists obtained from previous studies.
4. Discusion section:
4.1. "In this study, to investigate gene regulatory mechanisms, specifically on inflammation and its corresponding stress (i.e., oxidative stress) of pterygium, we utilized RNA-seq datasets of pterygium patients, our Asian pterygium and the European pterygium."
- This sentence can be change to: "In this study, we investigated gene regulatory mechanisms, specifically focusing on inflammation and its corresponding stress (i.e., oxidative stress) in pterygium, utilizing RNA-seq datasets from pterygium patients, including both Asian and European cohorts."
4.2. Consider rephrasing this sentence in order to clarify it: "The analyses from multiple point of views on the pterygium encompassed recognizing signaling pathways linked to current databases through GO analysis and GSEA, identifying potential gene regulation changes of core inflammatory gene list, estimating activity levels of core inflammatory functional modules, and computing correlativity levels between functional modules."
4.3. Consider rephrasing this sentence in order to clarify it: "ISR, intimately associated with NF-kB, contributes to the transcription of proinflammatory genes, and also upon activation, ISR leads to the secretion of inflammatory cytokines fostering communication between cells during local inflammation. Among the commonly upregulated genes in the corresponding modules, GSTA1 played a role in detoxification of products of oxidative stress [58]."
Reviewer 2 Report
Comments and Suggestions for Authors
The authors showed great study that they investigated the intrinsic gene regulatory mechanisms closely linked to the inflammatory reactions of pterygium. But I felt the contents of the authors study is too complicated. So I recommend the authors to add things in clinical situations. I have a few minor comments to be addressed.
1. Basically, Ophthalmologist will consider how pterygium will advance or influence cornel astigmatism in clinic situations. So, authors should describe how intrinsic gene and inflammatory reactions will correlate the progression of the pterygium.
2. Conclusion or discussion part, please speculate how authors think about pterygium progression in clinic situations. For example, through the authors results, anti-inflammatory eye drops will affect inflammatory pathway.
